# Development and Validation of Early Alert Model for Diabetes Mellitus–Tuberculosis Comorbidity

**DOI:** 10.3390/microorganisms13040919

**Published:** 2025-04-16

**Authors:** Zhaoyang Ye, Guangliang Bai, Ling Yang, Li Zhuang, Linsheng Li, Yufeng Li, Ruizi Ni, Yajing An, Liang Wang, Wenping Gong

**Affiliations:** 1Institute of Tuberculosis, Senior Department of Tuberculosis, The Eighth Medical Center of PLA General Hospital, Beijing 100091, China; yzy1997@mail.sdufe.edu.cn; 2Graduate School, Hebei North University, Zhangjiakou 075000, China; yl2015@mail.sdufe.edu.cn (L.Y.); zl199617@mail.sdufe.edu.cn (L.Z.); lls1989@mail.sdufe.edu.cn (L.L.); liyufeng@stu.sqxy.edu.cn (Y.L.); ruizini@stu.sdp.edu.cn (R.N.); anyajing@stu.sdp.edu.cn (Y.A.); 3Department of Geriatrics, The Eighth Medical Center of PLA General Hospital, Beijing 100091, China; 4Department of Clinical Laboratory, The Eighth Medical Center of PLA General Hospital, Beijing 100091, China; baigl1128@163.com

**Keywords:** diabetes mellitus, tuberculosis, immune-related biomarkers, machine learning, early alert model

## Abstract

Diabetes mellitus (DM) and tuberculosis (TB) are two global health challenges that significantly impact population health, with DM increasing susceptibility to TB infections. However, early risk prediction methods for DM patients complicated with TB (DM–TB) are lacking. This study mined transcriptome data of DM–TB patients from the GEO database (GSE181143 and GSE114192) and used differential analysis, weighted gene co-expression network analysis (WGCNA), intersecting immune databases, combined with ten machine learning algorithms, to identify immune biomarkers associated with DM–TB. An early alert model for DM–TB was constructed based on the identified core differentially expressed genes (DEGs) and validated through a prospective cohort study and reverse transcription quantitative real-time polymerase chain reaction (RT-qPCR) for gene expression levels. Furthermore, we performed a detailed immune status analysis of DM–TB patients using the CIBERSORT algorithm. We identified 1090 DEGs associated with DM–TB and further pinpointed CETP (cholesteryl ester transfer protein) (AUC = 0.804, CI: 0.744–0.864), TYROBP (TYRO protein tyrosine kinase binding protein) (AUC = 0.810, CI: 0.752–0.867), and SECTM1 (secreted and transmembrane protein 1) (AUC = 0.811, CI: 0.757–0.864) as immune-related biomarkers for DM–TB patients. An early alert model was developed based on these three genes (AUC = 0.86, CI: 0.813–0.907), with a sensitivity of 0.80829 and a specificity of 0.75758 at a Youden index of 0.56587. External validation using the GSE114192 dataset showed an AUC of 0.901 (CI: 0.847–0.955). Population cohort research and RT-qPCR verified the expression levels of these three genes, demonstrating consistency with trends seen in the training set. KEGG enrichment analysis revealed that NF-κB and MAPK signaling pathways play crucial roles in the DM–TB pathogenic mechanism, and immune infiltration analysis showed significant suppression of certain adaptive immune cells and activation of inflammatory cells in DM–TB patients. This study identified three potential immune-related biomarkers for DM–TB, and the constructed risk assessment model demonstrated significant predictive efficiency, providing an early screening strategy for DM–TB.

## 1. Introduction

Diabetes mellitus (DM) and tuberculosis (TB) are two significant public health challenges globally, each contributing substantially to morbidity and mortality rates. DM, characterized by chronic hyperglycemia due to defects in insulin secretion, insulin action, or both, has reached epidemic proportions, affecting approximately 589 million adults worldwide in 2024, with projections indicating a rise to 853 million by 2050 [1]. The disease is associated with various complications, including cardiovascular diseases, nephropathy, retinopathy, and neuropathy, which further exacerbate the healthcare burden and diminish patients’ quality of life [2,3]. On the other hand, TB, caused by *Mycobacterium tuberculosis* (MTB), remains a leading infectious disease worldwide, accounting for approximately 1.25 million deaths in 2023 alone [4]. The coexistence of DM and TB (DM–TB) poses a unique challenge as diabetes significantly increases susceptibility to TB infection and worsens treatment outcomes [3].

DM stands as a primary risk factor for TB [5]. Studies have shown that the incidence of TB among DM patients is much higher than in non-diabetic individuals [6,7]. This is primarily attributed to DM weakening the immune system, thereby reducing the body’s resistance to MTB [3,8,9,10]. Furthermore, the hyperglycemic state in DM patients creates favorable conditions for MTB survival and reproduction, further increasing the risk of infection [11,12,13,14]. When DM patients contract MTB, the disease course often becomes more complicated, leading to exacerbated symptoms, higher TB recurrence rates, and increased mortality [15,16].

Therefore, early warning, diagnosis, and interventional treatment for DM patients at risk of developing TB are crucial strategies to reduce DM–TB incidence and mortality rates. The booming development of artificial intelligence and multi-omics technologies provides new avenues to address this challenge [17,18]. This study focuses on early risk prediction for patients with DM–TB. Through deep mining of transcriptome data from the GEO database, combined with weighted gene co-expression network analysis (WGCNA) and ten machine learning algorithms, we integrate bioinformatics methods to identify immune biomarkers associated with DM–TB. Simultaneously, we construct an early alert model using these immune biomarkers and validate it via GEO datasets, prospective cohort studies, and reverse transcription quantitative real-time polymerase chain reaction (RT-qPCR). Considering the pivotal role of the immune system in DM–TB onset and progression, we not only identify disease-related immune biomarkers but also uncover vital immune signaling pathways such as NF-kappa B and MAPK. Additionally, immune infiltration analysis reveals significant suppression of certain adaptive immune cells and activation of inflammatory cells in DM–TB patients. These novel findings offer potential immunological targets for early detection and screening of DM–TB patients, paving the way for new treatment and prevention strategies. Through this research, we aspire to provide robust support for TB risk management among DM patients, ultimately improving their quality of life and survival rates.

## 2. Materials and Methods

### 2.1. Data Acquisition and Initial Processing

A comprehensive search within the Gene Expression Omnibus (GEO) repository was executed, utilizing the search terms “diabetes mellitus” and “tuberculosis”. This yielded a corpus of 99 datasets. Subsequently, a meticulous screening process was applied, informed by stringent inclusion criteria: (1) datasets must encompass microarray or RNA-sequencing technologies for gene expression profiling; (2) human mRNA expression data must be present; (3) all samples must be derived from whole blood; (4) the experimental design must incorporate a case-control framework, including distinct groups for DM, DM–TB, and healthy controls (HCs), each comprising a minimum of three samples. Following this stringent selection, two datasets, GSE181143 and GSE114192, were identified for further scrutiny. GSE181143 was earmarked as the training dataset, with GSE114192 serving as the validation dataset (refer to Table 1 and Figure 1 for details).

### 2.2. Differential Gene Expression Analysis

The raw count data from the curated datasets were procured, encompassing a total of 39,377 genes within the training dataset. To pinpoint genes that are distinctively expressed in DM–TB patients relative to the broader DM demographic, a differential gene expression analysis was conducted employing the R package “DESeq2” (R version 4.4.1). This analysis was preceded by background correction and data normalization to ensure the comparability and integrity of the data [19,20]. Genes exhibiting a fold change of at least 1.5 and a *p*-value less than 0.05 were deemed to be significantly differentially expressed between DM–TB patients and control samples. The “ggplot2” package (R version 4.4.1) was leveraged to graphically represent the findings in the form of a volcano plot, while the gene expression levels and correlation analyses were depicted using GraphPad Prism software (Version 10.0.0, San Diego, CA, USA).

### 2.3. Functional Enrichment Analysis

To elucidate the potential roles of the differentially expressed genes (DEGs) in the pathogenesis of DM–TB, both Gene Ontology (GO) and Kyoto Encyclopedia of Genes and Genomes (KEGG) pathway analyses were conducted. The GO analysis was designed to identify significant enrichment of these genes across biological processes, cellular components, and molecular functions. Conversely, KEGG pathway analysis aimed to delineate the involvement of these genes in intricate metabolic pathways and signal transduction networks, thereby offering a holistic view of gene functions and the key biological regulatory networks implicated in DM–TB [21,22].

### 2.4. Weighted Gene Co-Expression Network Analysis (WGCNA)

To further explore the interplay between DEGs and DM–TB, and to identify pivotal genes, the WGCNA approach was implemented [23,24]. This methodology identifies genes that are tightly co-expressed and aids in uncovering genes that are centrally associated with phenotypic traits. The “WGCNA” R package was utilized to discern gene modules pertinent to DM–TB [25]. A soft thresholding technique was employed to construct an apt weighted gene co-expression network, with the optimal soft threshold parameter β = 8 being ascertained by assessing network scale-free topology across a range of β values. Genes with analogous expression patterns were clustered into distinct modules predicated on topological overlap measures, employing hierarchical clustering in conjunction with dynamic tree-cutting methods [26]. The module resolution and stability were calibrated by setting a minimum module size of 30 and adjusting the sensitivity parameter to 3. Modules with a distance threshold of less than 0.25 were consolidated to mitigate redundancy and to enhance biological relevance.

### 2.5. Identification of Immune-Related Genes Utilizing ImmPort

Given the intricate interplay between the immune system and the progression of DM–TB, deciphering the underlying immune regulatory mechanisms is imperative. To this end, we accessed a comprehensive dataset of immune-related genes from the ImmPort database (https://www.immport.org/shared; accessed on 5 October 2024) [27]. This authoritative immunology repository offers high-fidelity immunology data, encompassing gene expression profiles, proteomics, and clinical information. An intersection analysis was performed between the immune-related gene dataset from ImmPort and the genes identified as pivotal by WGCNA, thereby pinpointing immune-related target genes specifically associated with DM–TB.

### 2.6. Machine Learning for Immune-Related Biomarker Selection

To pinpoint genes that exert a significant influence on the immune response in DM–TB, an array of machine learning algorithms was employed. Notably, the Least Absolute Shrinkage and Selection Operator (LASSO) was selected for its prowess in variable selection and regularization; Random Forest (RF) for its aptitude in non-linear modeling and resistance to overfitting; Support Vector Machine (SVM) for its proficiency in identifying optimal hyperplanes in high-dimensional spaces; Gradient Boosting for its iterative model refinement to minimize prediction errors; K-Nearest Neighbors for its straightforward classification predicated on similarity metrics; Decision Tree for its interpretable classification or regression models; Generalized Linear Model as a conventional tool for statistical modeling across diverse datasets; Neural Network for emulating the neural mechanisms of the human brain; XGBoost for its efficiency and scalability in Gradient Boosting; and C5.0 for its enhanced Decision Tree algorithm tailored for extensive datasets. These algorithms were executed using respective R packages, including glmnet, gbm, randomForest, e1071, caret, rpart, neuralnet, xgboost, and C50. Each algorithm independently appraised the significance of genes and ranked them accordingly. The top 10 genes from each algorithm were shortlisted as candidates.

To streamline the list and identify core genes, we focused on those that were consistently highlighted by multiple machine learning algorithms. These genes, which were deemed highly significant across various algorithms, were postulated as potential biomarkers for the immune response in DM–TB. The “UpSet” R package (R version 4.4.1) was utilized to generate an UpSet plot, which not only depicted the number of genes selected by each algorithm but also revealed the extent of concordance among different algorithms through set intersections and unions.

### 2.7. Prospective Cohort and Transcriptome Sequencing

To substantiate the expression patterns of the identified key differentially expressed genes within a real-world cohort, a prospective cohort study was initiated. Participants were recruited for the HCs (*n* = 10), DM (*n* = 10), and DM–TB (*n* = 10) groups. This study received approval from the Medical Ethics Committee of the Eighth Medical Center of PLA General Hospital (ethical approval number: 3092023122013297234), with all participants providing informed consent. Peripheral blood (5 mL) was collected from each participant to isolate peripheral blood mononuclear cells (PBMCs) and to extract total RNA. Following RNA quality assessment, small RNA libraries were constructed using the Illumina NovaSeq 6000 sequencing platform, and sequencing was conducted.

Inclusion criteria for the HC group included no history of TB exposure or previous TB diagnosis, normoglycemia, normal chest radiographs, human immunodeficiency virus (HIV) negative status, and age ≥ 12 years. Exclusion criteria encompassed previous residence or travel to high-risk TB areas, employment in TB-related medical or research fields, previous TB diagnosis or residual lung lesions, allergic reactions to IGRA testing, and the presence of DM or HIV infection.

Inclusion criteria for the DM group were fasting blood glucose (FBG) ≥ 7.0 mmol/L and glycated hemoglobin level ≥ 6.5%. Exclusion criteria for the DM group included HIV infection, granulocytopenia, autoimmune diseases, severe liver/kidney diseases, malignancies, immunomodulators use, and pregnancy.

Inclusion criteria for the DM–TB group satisfied both the aforementioned DM criteria and TB criteria. TB diagnosis adhered to the “Diagnostic Criteria for Pulmonary Tuberculosis WS288-2017”, encompassing patients with tuberculosis of the lungs, trachea, bronchi, or pleura [28,29]. Diagnostic evidence included etiological examination, epidemiological history, clinical manifestations, chest imaging, relevant auxiliary examinations, and differential diagnosis. A confirmed diagnosis necessitated positive etiological or pathological results. Exclusion criteria included the use of glucocorticoids, compromised immune function (e.g., HIV infection, post-organ transplantation, autoimmune diseases), malnutrition, and age < 12 years.

### 2.8. Retrospective Cohort and RT-qPCR

To corroborate the reliability of our bioinformatics predictions, RT-qPCR was performed on a retrospective cohort of 102 participants from the Eighth Medical Center of PLA General Hospital. This cohort comprised 36 DM–TB patients, 36 DM patients, and 30 HCs. Inclusion and exclusion criteria mirrored those previously described. TB patients were diagnosed according to the “Diagnostic Criteria for Pulmonary Tuberculosis WS288-2017”, while DM patients were diagnosed based on criteria from the American Diabetes Association. Participants outside the age range of 18 to 60 years, as well as those with cancer, immune system diseases, or other lung diseases, were excluded.

Total RNA was extracted from whole blood samples using the IVD Pure Total RNA Extraction Kit for Blood (IVDShow Biotechnology Huailai Co., Ltd., Huailai, Hebei, China) following the manufacturer’s protocol. Reverse transcription was conducted using the FastKing gDNA Dispelling RT SuperMix Kit (Tiangen Biotech Co., Ltd., Beijing, China) with incubation at 42 °C for 15 min followed by 95 °C for 3 min. Samples were diluted with RNase-free water. RT-qPCR of CETP (cholesteryl ester transfer protein), TYROBP (TYRO protein tyrosine kinase binding protein), and SECTM1 (secreted and transmembrane protein 1) was carried out using the FastReal qPCR Pre-210 Mix (SYBR Green) Kit (Tiangen Biotech Co., Ltd., Beijing, China) and the Roche 480 system. Reaction conditions included initial denaturation (95 °C, 2 min), followed by 40 cycles of denaturation (95 °C, 5 s), annealing, and extension (60 °C, 30 s). Glyceraldehyde-3-phosphate dehydrogenase (GAPDH) served as an internal control for amplification. Relative expression levels were quantified using the 2^−ΔΔCt^ method. Primer sequences utilized in this experiment are detailed in Table 2.

### 2.9. CIBERSORT Analysis of Immune Cell Infiltration

Immune cells are pivotal in the immune system, including pathogen identification and elimination, immune response regulation, and participation in inflammatory reactions. To explore shifts in immune cell abundance in the DM–TB state, the CIBERSORT algorithm was employed. This algorithm applies linear support vector regression to deconvolute gene expression profiles, estimating the prevalence of immune cells in samples derived from RNA sequencing data [30,31]. By conducting an in-depth analysis of the integrated gene expression matrix, we quantified the relative proportions of various immune cell types in the DM–TB and DM groups. The immune cell composition of the two groups was visualized using stacked bar charts generated with GraphPad Prism software and evaluated for changes in immune cell proportions using the Mann–Whitney test. Additionally, to investigate the relationship between the expression levels of immune-related biomarkers and shifts in immune cell proportions, we performed correlation analyses, uncovering potential associations between these biomarkers and specific immune cell types.

### 2.10. Statistical Analysis

Statistical analyses were conducted using GraphPad Prism software (Version 10.0.0, San Diego, CA, USA). For the comparison of gene expression levels and RT-qPCR relative expression data among DM–TB, DM, and HC groups, either one-way ANOVA or the non-parametric Kruskal–Wallis test was applied, contingent upon the normality and homogeneity of variances of the data. Furthermore, we constructed early alert models predicated on logistic regression analysis and evaluated the predictive performance of these models using Receiver Operating Characteristic (ROC) curves and calibration curves. A significance level of 0.05 was established for all statistical tests.

## 3. Results

### 3.1. Comparison of Clinical Characteristics Among (HCs, DM, and DM–TB Cohorts from Brazil and India (GSE181143 Dataset)

The demographic and clinical characteristics of the study participants in data set (GSE181143) are shown in Table 3. Comparative data between Brazilian and Indian cohorts were derived from a previous study by Caian L Vinhaes et al. [32], with additional analyses performed for the current context. Key findings include Brazilian healthy controls exhibited higher smoking (33.3% vs. 3.3%, *p* = 0.004) and alcohol use (86.7% vs. 18.4%, *p* < 0.001) than Indians, with no differences in age, BMI, sex, or HbA1c. In DM groups, Brazilians had higher BMI (30.3 vs. 25.4 kg/m^2^, *p* < 0.001), while Indians showed worse glycemic control (HbA1c 9.4% vs. 6.1%, *p* < 0.001) and lower alcohol use (22.5% vs. 93.4%, *p* < 0.001). Among DM–TB patients, Indians had higher cavitary TB prevalence (65% vs. 29%, *p* < 0.001), elevated HbA1c (11.7% vs. 8.5%, *p* = 0.001), and greater metformin (87% vs. 19.4%, *p* < 0.001) and statin use (22.5% vs. NA). Brazilians in DM–TB groups maintained higher smoking (38.7% vs. 15%, *p* = 0.02) and alcohol rates (90.3% vs. 12.5%, *p* < 0.001). Indian DM/DM–TB cohorts consistently demonstrated poorer glycemic control (*p* < 0.01).

### 3.2. Identification of DEGs in DM–TB Patients

Our analysis of the GSE181143 dataset from the GEO database, comprising 193 DM–TB patients, 66 DM patients, and 89 HCs, identified a total of 39,374 genes. Employing R language analysis with DM patients as the control group, we discerned 1090 unique DEGs in DM–TB patients, of which 573 were upregulated and 517 were significantly downregulated. These gene expression alterations were graphically represented through volcano plots and heatmaps (Figure 2).

In the volcano plot (Figure 2A), genes were ranked based on fold change, revealing the top 10 upregulated genes—ANKRD22, GBP1P1, BATF2, CD177, LIPM, GBP6, SERPING1, LINC02528, C1QC, and FCGR1CP—which are implicated in various processes including myeloid leukocyte activation and defense response to pathogens, crucial for host resistance to MTB infection. The top 10 downregulated genes—LOC102724621, KRT7, COL1A1, FN1, TTBK1, XIST, ATP2B3, TSIX, LOC107987293, and CDC27P11—are associated with key biological processes such as extracellular matrix stability, cell adhesion, and signal transduction, and their downregulation may reflect altered tissue structure and impaired repair mechanisms in DM–TB pathology.

Furthermore, we presented the top 45 genes with the highest fold changes in both groups, including 32 upregulated and 13 downregulated genes (Figure 2B–D). Compared to the DM group, the significant increase in gene expression levels in the DM–TB group (Figure 2B) suggests that these genes may play important roles in the development and progression of the comorbidity. Gene correlation analysis revealed three positively correlated red modules in the DM–TB group compared to the DM group (Figure 2D). The upper left and middle modules correspond to upregulated genes, indicating elevated expression levels in DM–TB and potential importance in the pathology. The lower right module represents downregulated genes, possibly indicating suppression or dysregulation in DM–TB (Figure 2D).

### 3.3. GO and KEGG Enrichment Analysis of DEGs in DM–TB Patients

Subsequently, we conducted GO enrichment analysis to explore the biological functions and potential pathogenesis implied by these DEGs. After screening for GO terms with a *p*-value less than 0.05, we ordered and interpreted them based on the number of involved genes (Figure 3). Within the BP category, we identified 453 significantly enriched GO terms covering key areas like neutrophil degranulation, activation, humoral immune response, negative regulation of immune system processes, defense response to bacteria, regulation of inflammatory response, positive regulation of cell adhesion and cytokine production, and the immune response-activating cell surface receptor signaling pathway. These findings not only reveal the complex immune response patterns in DM–TB but also highlight the important roles of specific cell types and their functions in disease progression (Figure 3A). The CC category revealed 81 enriched terms, focusing on specific granules, collagen-containing extracellular matrix, and lumen of secretory granules, structures critical for immune cell function and signaling (Figure 3B). In the MF category, we found 84 significantly enriched GO terms, including actin binding, ion channel activity, endopeptidase activity, channel activity, passive transmembrane transporter activity, cation channel activity, glycosaminoglycan binding, and gated channel activity. These diverse and interrelated functions form the basis of the biological roles played by the DEGs in DM–TB (Figure 3C).

Furthermore, KEGG enrichment analysis confirmed strong connections between these DEGs and biological pathways highly relevant to cancer transcription misregulation, human papillomavirus infection, NOD-like receptor signaling pathway, and systemic lupus erythematosus (Figure 4A). Notably, the NOD-like receptor signaling pathway (hsa04621) was significantly enriched with 13 upregulated DEGs (Appendix A). Further analysis indicated close associations between the upregulation of these genes and the NF-κB and MAPK signaling pathways, key regulators of inflammation and immune responses in DM–TB (Figure 4B).

### 3.4. Identification of Core Genes in DM–TB Using WGCNA

To delve deeper into the core genes within the DEG set, we constructed a co-expression network using WGCNA. Network topological analysis determined an optimal soft thresholding power of β = 8, where the scale-free network topology fit index R^2^ reached 0.87, indicating a robust network suitable for further module detection analysis. The network’s average connectivity demonstrated stability (Figure 5A,B). Subsequent module merging using a MEDissThres of 0.25 identified three key co-expression modules (Figure 5C). Detailed investigation into the relationship between modules and disease state revealed that the blue (R = 0.41, *p* < 0.0001) and gray modules (R = 0.45, *p* < 0.0001) were significantly positively correlated with the DM–TB group while negatively correlated with the DM group (R= −0.41, *p* < 0.0001; R = 0.45, *p* < 0.0001), suggesting these modules may play positive regulatory roles in DM–TB pathogenesis. Conversely, the turquoise module exhibited a weak negative correlation with the DM–TB group (R= −0.12, *p* = 0.05) and a positive correlation with the DM group, indicating potential downregulation in DM–TB patients, possibly related to immune suppression (Figure 5D).

To precisely pinpoint the key genes most closely associated with DM–TB, we implemented stringent selection criteria (MM threshold of 0.8, GS threshold of 0.1), ultimately identifying 181 core genes across the three modules. These genes not only demonstrated significant differential expression but also are likely critical drivers of DM–TB disease progression.

### 3.5. ImmPort Identification of Immune-Related Genes in DM–TB

Focusing on immune-related genes, we further employed the ImmPort database’s 1793 immune genes as references for cross-referencing with WGCNA-identified key module genes. This approach successfully helped in recognizing 20 overlapping immune-related genes (Figure 5E). These genes may play crucial roles in regulating DM–TB immune responses, offering potential as diagnostic biomarkers and new perspectives for understanding the immune pathogenesis of DM–TB.

### 3.6. Machine Learning Identification of Candidate Biomarkers for DM–TB

In this study, to explore the potential immune-related biomarkers among the 20 genes, we applied ten cutting-edge machine learning algorithms for data analysis. Initially, LASSO selected 11 genes as preliminary candidates (Figure 6A,B). Subsequently, SVM further pinpointed 16 candidate genes with outstanding performance in the model (Figure 6C). To ensure result reliability, we incorporated RF, GB, KNN, Decision Tree, GLM, NNET, XGBoost, and C5.0 algorithms, selecting the top 10 high-priority candidate genes from each (Figure 6D–L). This series of operations ensured comprehensive candidate gene coverage and enhanced reliability through multi-algorithm validation.

To identify the core intersection among these candidate genes, we used UpSet analysis to demonstrate the overlap among the ten machine learning algorithms (Figure 7A). Surprisingly, CETP, TYROBP, and SECTM1 exhibited prominent performance across most or all selections, becoming three common intersecting genes. Finally, to further observe expression changes of these immune-related genes in HCs, DM, and DM–TB groups, we compared their expressions in the three groups. The results showed significant elevation in gene expression in the DM–TB group compared to the DM and HCs groups (Figure 7B), suggesting increased immune cell activation in DM–TB patients, which might exacerbate pathological tissue damage.

### 3.7. Logistic Regression Prediction of DM–TB Risk: Construction of a Nomogram

For more precise early risk alerts for DM progressing into DM–TB, we developed an early alert model based on the identified three core genes (CETP, TYROBP, and SECTM1). This model, developed using binary logistic regression analysis, indicated that CETP (OR: 1.051, CI: 1.033–1.068; *p* < 0.0001), TYROBP (OR: 1.004, CI: 1.003–1.006; *p* < 0.0001), and SECTM1 (OR: 1.005, CI: 1.003–1.007; *p* < 0.0001) are important differential genes between DM–TB and diabetes (Table 4). The model formula is as follows: logit (*p*) = −2.111 + 0.559 ∗ CETP + 0.005 ∗ TYROBP + 0.017 ∗ SECTM1. To visually demonstrate the predictive effectiveness of this diagnostic model, we visualized the model using a nomogram (Figure 7C).

Next, to evaluate the predictive accuracy of this early risk alert model for DM–TB, we used the ROC curve for performance validation. The ROC analysis revealed AUCs of 0.804 (CI: 0.744–0.864) for CETP, 0.810 (CI: 0.752–0.867) for TYROBP, and 0.811 (CI: 0.757–0.864) for SECTM1 (Figure 8A). The overall model AUC was 0.86 (CI: 0.813–0.907), with sensitivity at 0.80829 and specificity at 0.75758 at a Youden index of 0.56587 (Figure 8B). These data indicate that the three-gene model’s early warning capability was significantly superior to single-gene models. Furthermore, for model accuracy assessment, we plotted a calibration curve, which showed the risk alert model had good accuracy (Figure 8C).

### 3.8. External Dataset and Population Cohort Validation of the DM–TB Early Risk Alert Model

Additionally, we used GSE114192 as a validation set to further verify the model’s applicability to external data. In this validation set, the early warning effectiveness of the three genes alone for DM–TB were AUC of 0.805 (CI: 0.725–0.885) for CETP, 0.833 (CI: 0.759–0.908) for TYROBP, and 0.892 (CI: 0.832–0.952) for SECTM1 (Figure 8D), with combined model warning ability increased to AUC = 0.901 (CI: 0.847–0.955) (Figure 8E). The calibration curve indicated good stability of the model (Figure 8F). However, gene expression trends in the validation set did not consistently match those in the training set. Factors affecting gene expression, such as the absence of patient background, disease control, recurrence, and treatment experiences, as well as sample origin from multiple countries, may contribute to these differences. Further clinical data and background information would be necessary for deeper analysis of gene expression variations across different populations.

To validate the expression of these core genes in real-world populations, we conducted a prospective cohort study involving 10 DM–TB patients, 10 DM patients, and 10 HCs. The results showed that the predictive performance of the three genes individually for DM–TB was as follows: CETP had an AUC of 0.77 (CI: 0.553–0.987), TYROBP had an AUC of 0.69 (CI: 0.432–0.948), and SECTM1 had an AUC of 0.87 (CI: 0.697–1.000) (Figure 9A). Figure 9B showed no significant difference in CETP between groups, though trends were consistent with the training set. TYROBP showed no significant difference between the DM and DM–TB groups but was significantly lower in the DM group compared to HCs (*p* = 0.0022, Figure 9C). SECTM1 expression was significantly higher in DM–TB than in the DM group (*p* = 0.0089) and significantly lower in the DM group compared to healthy controls (*p* = 0.0144, Figure 9D).

Finally, validation by RT-qPCR in a retrospective cohort confirmed a similar trend: there was no significant difference in CETP in the DM–TB group compared with the DM group, but there was still an upward trend, and TYROBP (*p* = 0.0357) and SECTM1 (*p* = 0.0171) were significantly higher than those in the DM group (Figure 9E–G). These findings indicate that these three core genes are potential biomarkers for the early risk prediction of DM–TB, with the constructed risk alert model based on them demonstrating good accuracy and offering a new method for early risk warnings.

### 3.9. Immune Infiltration Analysis in DM–TB and DM Patients

The immune response plays a crucial role in the development and progression of DM–TB. To gain insights into the immune status of DM–TB patients, we performed immune infiltration analysis using the CIBERSORT algorithm. Our results showed that neutrophils, monocytes, CD4^+^ T cells, CD8^+^ T cells, NK cells, and B cells were the most abundant immune cells within both DM–TB and DM groups. Further analysis revealed a noticeable reduction in the proportion of adaptive immune cells and an increase in inflammatory cells in the DM–TB group (Figure 10A). Specifically, the proportion of naive B cells (*p* = 0.000152), memory B cells (*p* = 0.000985), plasma cells (*p* = 0.000119), CD8^+^ T cells (*p* = 0.000119), resting CD4^+^ memory T cells (*p* < 0.0001), and activated dendritic cells (*p* = 0.036216) were significantly lower in the DM–TB group compared to the DM group (Figure 10B). This suggests possible immune suppression in DM–TB patients, leading to reduced infection response capabilities. Concurrently, the proportions of monocytes (*p* = 0.004620), resting mast cells (*p* = 0.036216), and neutrophils (*p* < 0.0001) were significantly elevated (Figure 10B), potentially indicating enhanced inflammatory responses and immune dysregulation in DM–TB patients.

Additionally, to further explore the relationship between the core genes CETP, TYROBP, SECTM1, and infiltrating immune cells, a correlation analysis was conducted. The results indicated that CETP was positively correlated with memory B cells (*p* < 0.001, R = 0.216) and resting CD4^+^ memory T cells (*p* = 0.010, R = 0.161) (Figure 10C); TYROBP was positively correlated with naive CD4^+^ T cells (*p* = 0.0153, R = 0.1506) (Figure 10C); SECTM1 was negatively correlated with naive CD4^+^ T cells (*p* = 0.0095, R = 0.1608) and macrophages M0 (*p* = 0.0474, R = −0.1233) (Figure 10C).

To further understand how these genes express in the DM–TB vs. DM contexts, we performed grouped correlation analyses between expression levels and infiltrating immune cells. In the DM–TB group, (1) CETP was negatively correlated with naive B cells, CD8^+^ T cells, resting CD4^+^ memory T cells, resting NK cells, and M0 macrophages while positively correlated with plasma cells, M1 macrophages, and neutrophils (Figure 10C); (2) TYROBP was negatively correlated with naive B cells, CD8^+^ T cells, naive CD4^+^ T cells, resting CD4^+^ memory T cells, resting NK cells, and eosinophils while positively correlated with plasma cells, activated CD4^+^ memory T cells, Tregs, γδ T cells, and neutrophils (Figure 10C); (3) SECTM1 was negatively correlated with naive B cells, CD8^+^ T cells, naive CD4^+^ T cells, resting CD4^+^ memory T cells, and resting NK cells while positively correlated with plasma cells, activated CD4^+^ memory T cells, Tregs, γδ T cells, activated dendritic cells, and neutrophils (Figure 10C).

In the DM group, (1) CETP was negatively correlated with naive B cells, naive CD4^+^ T cells, activated CD4^+^ memory T cells, and eosinophils (Figure 10C); (2) TYROBP was negatively correlated with naive CD4^+^ T cells and resting CD4^+^ memory T cells while positively correlated with plasma cells and activated CD4^+^ memory T cells (Figure 10C); (3) SECTM1 was negatively correlated with naive CD4^+^ T cells while positively correlated with activated CD4^+^ memory T cells (Figure 10C).

These comprehensive findings suggest an increased correlation between gene expression levels and immune cell proportions in the DM–TB group compared to the DM group, predominantly showing inverse correlations. This may indicate significant immune suppression in DM–TB patients, leading to reduced pathogen defense capabilities and potentially contributing to disease progression and exacerbation.

## 4. Discussion

DM and TB are burgeoning global health challenges, with DM patients exhibiting a more than threefold increased risk of TB due to compromised immune function. This comorbidity aggravates patients’ conditions, escalating drug resistance, recurrence rates, and mortality. Thus, early risk warning and intervention management for DM–TB are of paramount importance. This study harnessed bioinformatics and artificial intelligence technologies to identify three immune biomarkers associated with DM–TB—CETP, TYROBP, and SECTM1—and constructed an early risk alert model, offering novel strategies for early screening and immune intervention in DM–TB patients.

Our analysis of transcriptomic data from the GEO database identified 1090 DEGs. GO enrichment analysis revealed these genes’ close association with the negative regulation of immune system processes, humoral immune response, inflammatory response regulation, cytokine production, and immune response activation on the cell surface receptor signaling pathways. KEGG enrichment analysis highlighted the NOD-like receptor signaling pathway (hsa04621), significantly enriching 13 upregulated DEGs. This pathway’s activation by NOD-like receptor proteins upon recognizing bacterial peptidoglycan components leads to the recruitment of downstream signaling molecules like RIP2, subsequently activating the NF-κB and MAPK signaling pathways [33], widely involved in immune and inflammatory responses, cell proliferation, survival, and apoptosis [34,35].

Studies suggest that these pathways are closely related to DM–TB pathogenesis. JYOTI RANI et al. identified disrupted pathways contributing to TB susceptibility in T2DM patients, implicating NF-κB [34]. Additionally, research indicates that NF-κB signaling inhibition through exercise can ameliorate pulmonary inflammation and reduce oxidative stress in DM patients [35]. Furthermore, Nuria Martinez et al. found significant upregulation of the MAPK pathway in T cells of hyperglycemic mice, potentially leading to pathological inflammation [36]. Other studies have also underscored the critical role of NF-κB and MAPK signaling pathways in TB immune regulation [37,38,39]. Notably, while the NF-κB/MAPK pathways are activated in DM–TB (Figure 4B), our data reveal significant reductions in adaptive immune cell populations, such as CD8+ T cells and memory B cells (Figure 10A). This apparent contradiction may reflect a state of immune exhaustion driven by chronic immune activation. Under persistent inflammatory stimuli (e.g., MTB infection combined with hyperglycemia), sustained NF-κB/MAPK hyperactivation may trigger negative feedback mechanisms [40], such as upregulating inhibitory receptors (e.g., PD-1 and CTLA-4) or Tregs, thereby suppressing effector T-cell function. Similar mechanisms are well documented in chronic viral infections and cancer [41,42,43].

WGCNA identified key gene modules, isolating 181 DEGs closely linked to DM–TB. Intersecting these genes with immunoregulatory genes from ImmPort yielded 20 target genes highly relevant to immune regulation. Various algorithms were employed to refine the candidate genes, ultimately identifying CETP, TYROBP, and SECTM1 as core genes. These core genes showed slightly downregulated expression in the DM population compared to healthy controls but were significantly overexpressed in the DM–TB group, implicating their involvement in the body’s response against MTB infection.

Based on these findings, we further analyzed the roles of the three core genes in the progression from DM to DM–TB. CETP, TYROBP, and SECTM1 are integral biomarkers in the pathogenesis of DM–TB, exerting a coordinated influence on disease progression and severity through their effects on lipid metabolism, immune cell activation, and IFN-γ production. CETP, a key regulator of lipoprotein metabolism [44,45], has been implicated in the pathogenesis of both DM and TB [46]. In the context of DM–TB, CETP’s role in lipid metabolism is particularly pertinent [44,45,46,47,48,49] as it modulates the intracellular lipid accumulation in macrophages post-MTB infection [50,51,52]. This process not only supplies nutrients to the bacillus but also constructs a protective milieu that shields MTB from host immune defenses. Our investigation of the CETP within the DM–TB cohort, relative to the DM and HC groups, revealed notably elevated CETP expression levels in the DM–TB group within the GEO training dataset. This observation suggests a potential exacerbation of lipid metabolic dysregulation in individuals with DM–TB. CETP expression did not exhibit statistically significant differences across the validation cohorts (GEO, population, and RT-qPCR), although a persistent upward trend was observed in the DM–TB group, indicating the presence of subclinical lipid dysregulation that warrants further investigation. The lack of significance in the population cohort and RT-qPCR analyses may be attributed to several factors. First, the relatively small sample size may have limited the statistical power to detect meaningful differences. Additionally, the use of whole blood RNA introduces potential limitations inherent in its heterogeneous cellular composition. Specifically, the presence of multiple cell types in whole blood can lead to signal dilution, particularly if CETP expression is predominantly observed in monocytes rather than lymphocytes, thereby masking the true biological changes.

TYROBP or DAP12 (DNAX activation protein 12 kDa) is a signaling transduction molecule that can activate antigen-presenting cells (APCs) like macrophages and dendritic cells by interacting with various cell surface receptors. Previous studies have associated DAP12 with delayed TB immune responses [53,54,55,56]. M Jeyanathan et al.’s research demonstrates that DAP12 upregulates IRAK-M, increasing IL-10, an immunosuppressive cytokine that inhibits APC activation, reducing NF-κB nuclear translocation [53,54]. These changes diminish APC activity, delaying Th1 cell responses in lymph nodes and lungs [57]. Besides delaying immune responses, DAP12 can also facilitate MTB immune evasion [58]. Ei’ichi Iizasa et al. found that DAP12 upregulation enhances TREM2 activity, which induces “permissive” macrophages against Mincle/FcRγ/CARD9-mediated immune responses, impairing macrophage bactericidal activity, thus promoting MTB survival [58]. DAP12 is also closely related to T2DM pathogenesis [59], forming a signaling complex with lipopolysaccharide (LPS), activating Toll-like receptor 4 (TLR4), triggering inflammatory cytokine storms, leading to insulin resistance and ultimately T2DM development [59,60]. Numerous studies have identified TYROBP as a biomarker for diabetic nephropathy and retinopathy [61,62,63,64]. Thus, our study suggests that TYROBP is closely tied to DM–TB, with potential as an immune biomarker.

SECTM1 is a member of the secreted and transmembrane protein family, with widespread expression in various cell types, particularly immune cells [45]. Research indicates that SECTM1 is a pivotal co-stimulatory ligand that promotes interferon-γ (IFN-γ) production and CD4 and CD8 T-cell proliferation [65]. Additionally, SECTM1 may enhance anti-CD28 signaling through the PI3K pathway to foster T-cell proliferation and activation [65]. In MTB infection immune responses, IFN-γ crucially activates immune cells, enhances antimicrobial capacity, and promotes cytokine production, aiding in effective MTB resistance [66]. Notably, studies employing multiomics approaches have shown that SECTM1 gene expression levels correlate positively with AFB grading in DM–TB patients’ sputum, indicating it as a marker associated with DM–TB pathophysiology [32]. Consistent with this, our results show significant SECTM1 upregulation in the DM–TB group, suggesting extensive immune cell activation likely linked to DM–TB-induced severe pathological damage. Thus, SECTM1 may serve as a potential marker for assessing DM–TB disease severity in the future.

Using the three core genes, we developed an early risk alert model for the DM–TB population, intending to provide early TB infection risk prediction for DM patients. In the GSE181143 dataset, the model’s AUC was 0.86, and it was 0.901 in the GSE114192 dataset. We observed that the combined core gene model had a significantly higher AUC than individual gene models, indicating enhanced predictive capability by integrating all three core genes. Additionally, gene expression levels verified by prospective cohort study and RT-qPCR verified by retrospective cohort confirmed trends consistent with the training set across groups, demonstrating the model’s predictive potential.

Furthermore, using the CIBERSORT algorithm, we conducted an in-depth analysis of the immune status in DM–TB patients. The results revealed significant changes in multiple immune cell types in the DM–TB group compared to the DM group. Specifically, the proportions of naïve B cells, memory B cells, plasma cells, CD8^+^ T cells, resting CD4^+^ memory T cells, and activated dendritic cells were notably decreased in the DM–TB group, indicating impaired immune responses. Previous studies found that hyperglycemia significantly reduces dendritic cell frequency [67], which, as effective APCs, could delay adaptive immune cell activation when reduced. Figure 10B demonstrates significant reductions in both innate and adaptive immune cell frequencies in DM–TB patients compared to DM-only patients. Research suggests that CD4^+^ T cells are central in eliminating MTB [68,69,70], where MTB infection should largely activate these cells. However, activated CD4^+^ memory T cells were significantly fewer than naïve CD4^+^ T cells and resting CD4^+^ memory T cells, with other T cell types also showing low levels. Although NK cells showed no significant differences between groups, activated NK cells were low, with more in a resting state. Macrophages play a vital role in TB infection, and our data indicated a low level of all three macrophage types (Figure 10B), suggesting immune suppression by MTB. Meanwhile, Figure 10B shows significant elevation of monocyte and neutrophil proportions, reflecting heightened inflammatory responses and immune imbalance in MTB response (Figure 10B). Further analysis revealed a significantly increased correlation between gene expression levels and immune cells in the DM–TB group compared to the DM group, predominantly negative correlations, suggesting more pronounced immune suppression in DM–TB patients, impairing pathogen defense.

The identified biomarkers (CETP, TYROBP, and SECTM1) hold translational potential for non-invasive TB screening in diabetic populations. Their detection via cost-effective RT-qPCR could enable early risk stratification in high-burden regions, reducing reliance on costly radiologic or pathogen-based diagnostics. By facilitating timely prophylaxis, this approach may curb hospitalization rates, mitigate drug resistance, and lower long-term healthcare costs. It is important to acknowledge the limitations inherent in our research. Firstly, the application of stringent inclusion criteria restricted the dataset to a single training set and a single validation set, thereby constraining the sample size available for the screening and identification of DEGs. This limitation may have impacted the generalizability of our findings. Secondly, fiscal limitations circumscribed our prospective cohort study to a modest sample size of ten volunteers per group, potentially affecting the statistical power of our observations. Third, our analysis of the biological functions of the three core genes—CETP, TYROBP, and SECTM1—was confined to the molecular level. We did not extend our investigation to include in-depth functional assessments using animal models, which could have provided additional insights into the genes’ roles in vivo. Fourth, the cohort consisted mainly of populations from specific regions (e.g., Brazil and India) and lacked validation in multiethnic groups. Genetic and environmental heterogeneity across populations may affect biomarker performance, necessitating external validation across populations. Fifth, direct clinical interpretation of their diagnostic or prognostic utility is hampered by the lack of clinical correlation analyses, such as correlations between biomarkers and TB severity or glycemic control. Finally, bulk RNA-seq of whole blood inherently masks cell-type-specific expression patterns due to the heterogeneous cellular composition of peripheral blood. Future studies should prioritize the integration of single-cell RNA sequencing (scRNA-seq) or fluorescence-activated cell sorting (FACS), which can pinpoint biomarker expression in disease-associated immune subpopulations (e.g., monocytes and neutrophils).

## 5. Conclusions

In conclusion, our comprehensive analysis has successfully identified key genes and biological markers associated with diabetes and tuberculosis co-morbidity. The integration of differential gene expression analysis, functional enrichment assessments, and machine learning methodologies has provided valuable insights into the underlying molecular mechanisms of DM–TB. These findings not only contribute to the existing knowledge of the disease interplay but also pave the way for future research aimed at early diagnosis and targeted therapeutic interventions. However, the study’s limited sample size necessitates validation in large-scale, multiethnic cohorts. Ultimately, the identification of specific biomarkers holds significant promise for improving clinical management and patient outcomes in DM–TB populations, underscoring the necessity for continued exploration in this critical area of study.

## Figures and Tables

**Figure 1 microorganisms-13-00919-f001:**
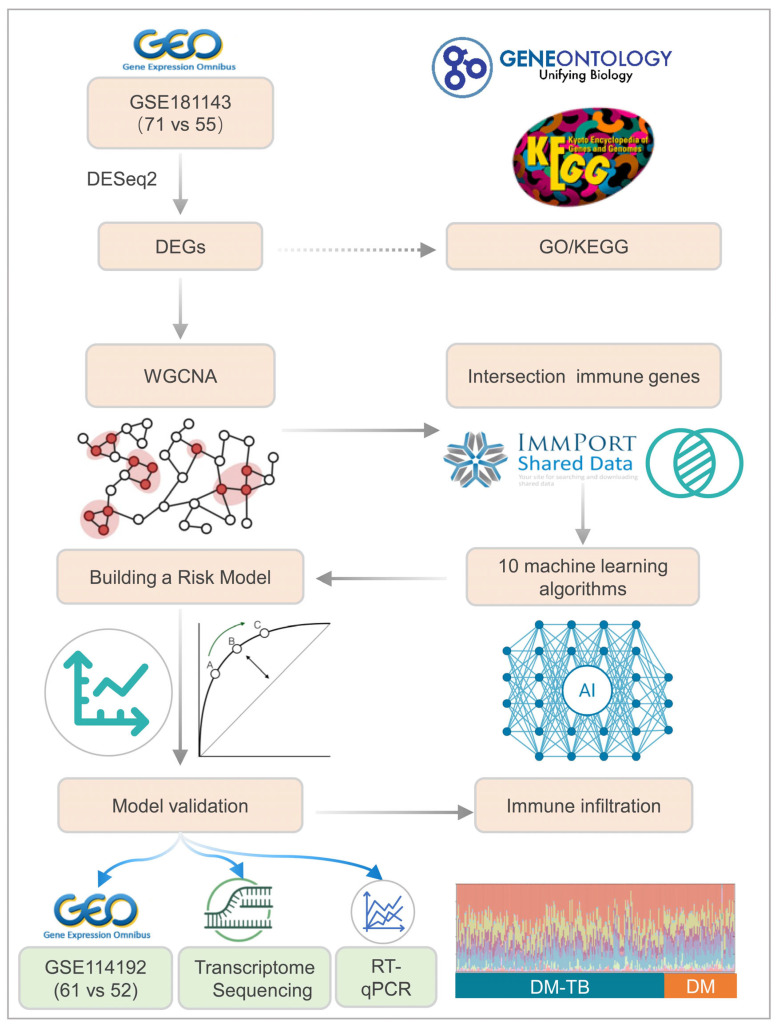
Flowchart of the construction and validation of the early risk alert model for DM–TB patients.

**Figure 2 microorganisms-13-00919-f002:**
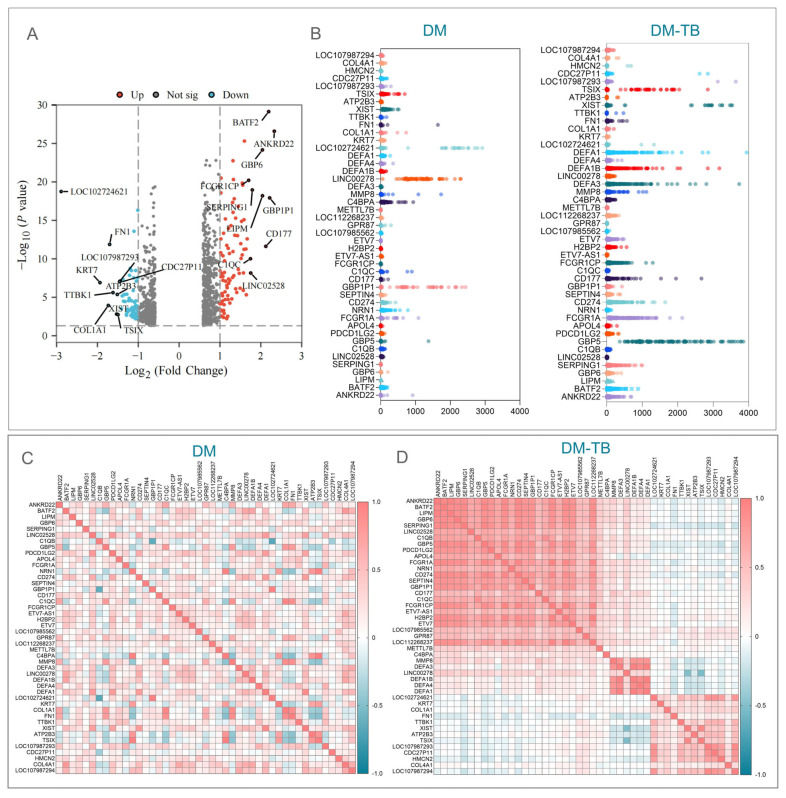
Differential analysis. (**A**) The volcano plot displays 1090 differentially expressed genes between the DM–TB and DM groups, with upregulated genes in red and downregulated genes in blue. (**B**) Expression levels of the top 45 genes with the highest differential fold change between the two groups are shown, with the top 13 being downregulated and the bottom 32 upregulated. (**C**,**D**) Correlation analysis of the 45 genes between the two groups, with the top 32 genes being upregulated and the bottom 13 downregulated.

**Figure 3 microorganisms-13-00919-f003:**
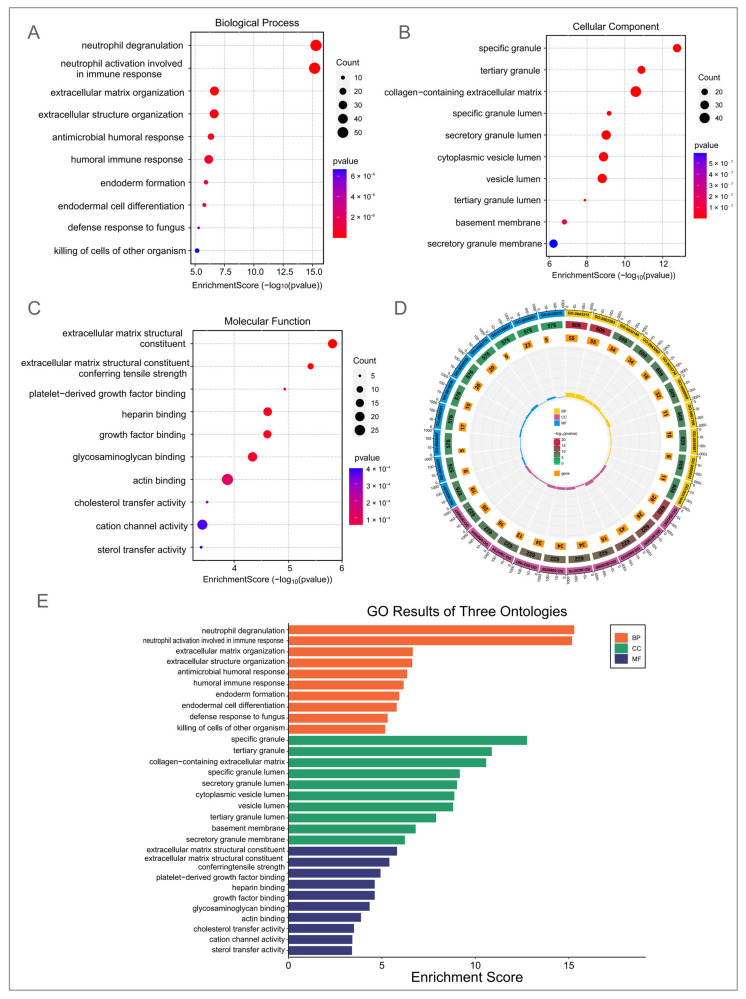
GO enrichment analysis of differentially expressed genes. (**A**) Enrichment analysis of biological processes highlighting significant processes related to the differentially expressed genes. (**B**) Enrichment analysis of molecular functions, emphasizing the roles of differentially expressed genes in specific functions. (**C**) Enrichment analysis of cellular components, revealing the distribution of differentially expressed genes across cellular compartments. (**D**) Circular visualization of GO analysis results, integrating enrichment across various functional categories. (**E**) GO enrichment results for three categories, ranked by significance scores.

**Figure 4 microorganisms-13-00919-f004:**
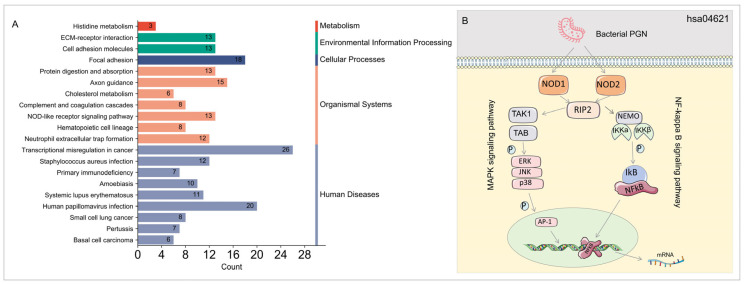
KEGG functional enrichment analysis of differentially expressed genes. (**A**) KEGG classification diagram. (**B**) NOD-like receptor signaling pathway (hsa04621). NLR proteins (NOD1 and NOD2) are activated by recognizing bacterial peptidoglycans. Upon activation, NLR recruits downstream signaling molecules RIP2, activating the NF-κB and MAPK pathways. Specifically, RIP2 activates the IκB activation complex (composed of NEMO, IKKα, and IKKβ), leading to IκB protein phosphorylation and degradation, releasing the NF-κB dimer, which translocates to the nucleus to promote target gene transcription. Additionally, RIP2 activates TAK1 and TAB, further leading to ERK, JNK, and p38 activation, phosphorylating nuclear AP-1, and promoting target gene transcription. Abbreviations: AP-1, activator protein 1; ERK, extracellular signal-regulated kinase; IκB, inhibitor of κB; IKKα/IKKβ, inhibitor of nuclear factor κB kinase subunit α/β; JNK, c-jun N-terminal kinase; MAPK, mitogen-activated protein kinase; NF-κB, nuclear factor kappa-light-chain-enhancer of activated B Cells; NOD1/NOD2, nucleotide-binding oligomerization domain-containing protein 1/2; NEMO, NF-κB essential modulator; RIP2, receptor-interacting protein 2; TAK1, TGF-β-activated kinase 1; TAB, TAK1-binding protein.

**Figure 5 microorganisms-13-00919-f005:**
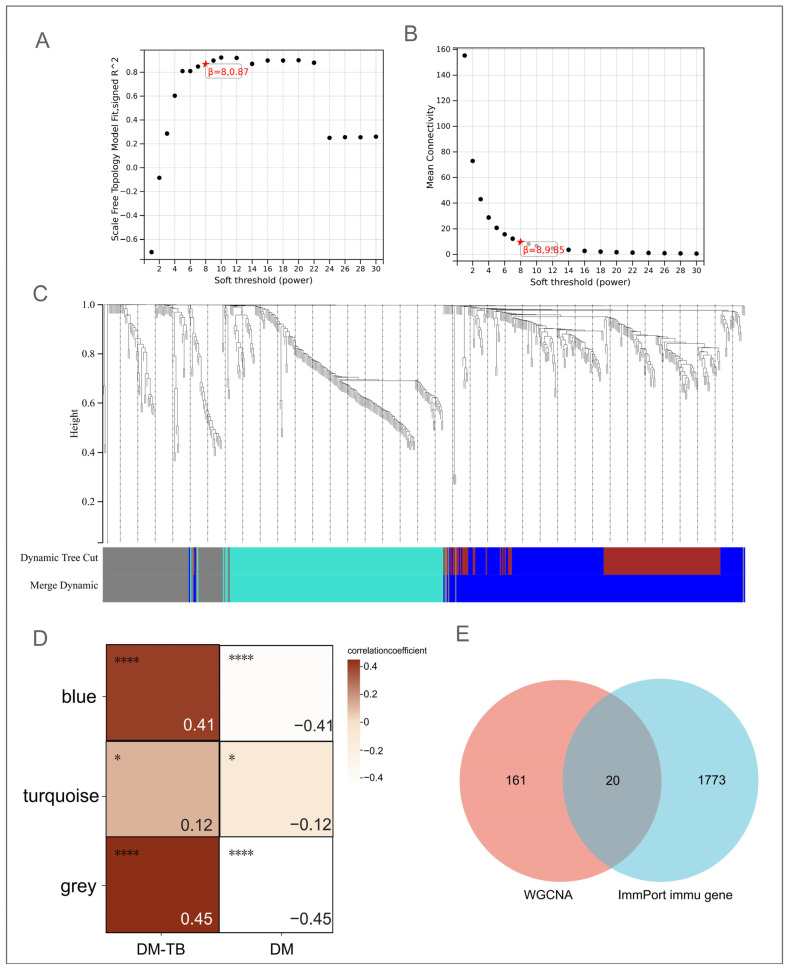
Weighted gene co-expression network analysis (WGCNA). (**A**,**B**) Scale independence and mean connectivity determine the optimal soft threshold as 8. (**C**) Clustering dendrogram of highly connected genes in key modules. (**D**) Relationship between modules and the two disease groups. (**E**) Intersection of the 181 core genes identified in WGCNA with 1793 immune genes from the ImmPort database, resulting in 20 immune-related core genes. *: indicate statistical significance: *: *p* < 0.05, ****: *p* < 0.0001.

**Figure 6 microorganisms-13-00919-f006:**
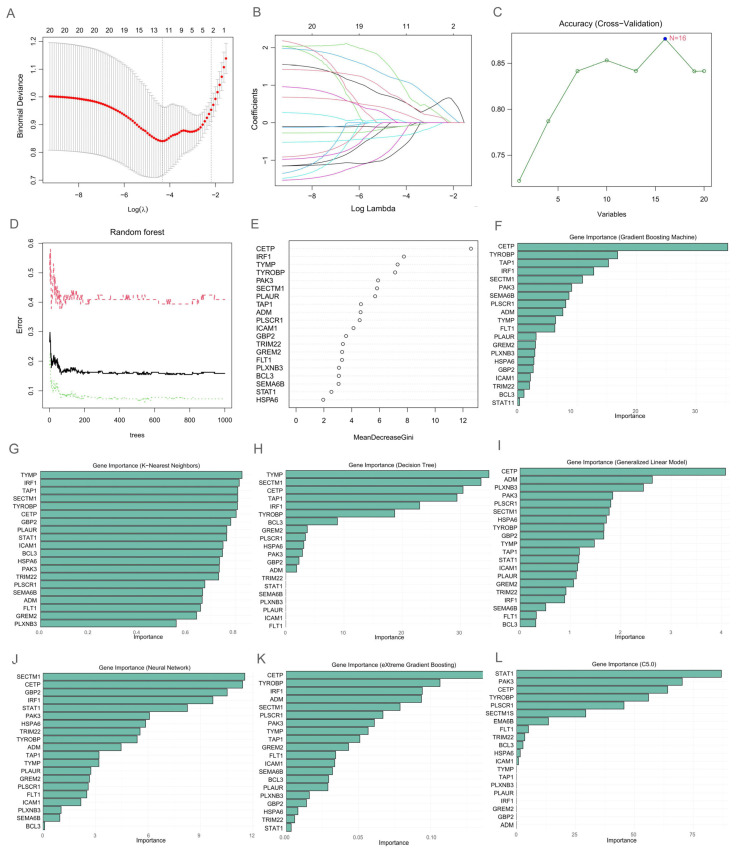
Machine learning selection of biomarkers. (**A**,**B**) LASSO. (**A**) Cross-validation error profile; (**B**) Coefficient shrinkage paths: Colored trajectories depict the standardized coefficients of predictors against log(λ); (**C**) SVM—Support Vector Machine; (**D**,**E**) RF—Random Forest; (**D**) Error comparison across random forest configurations; (**E**) Feature importance ranking in random forest; (**F**) GB—Gradient Boosting; (**G**) KNN—K-Nearest Neighbors; (**H**) Decision Tree; (**I**) GLM—Generalized Linear Model; (**J**) NNET—Neural Network; (**K**) XGBoost—eXtreme Gradient Boosting; (**L**) C5.0—C5.0 Decision Tree.

**Figure 7 microorganisms-13-00919-f007:**
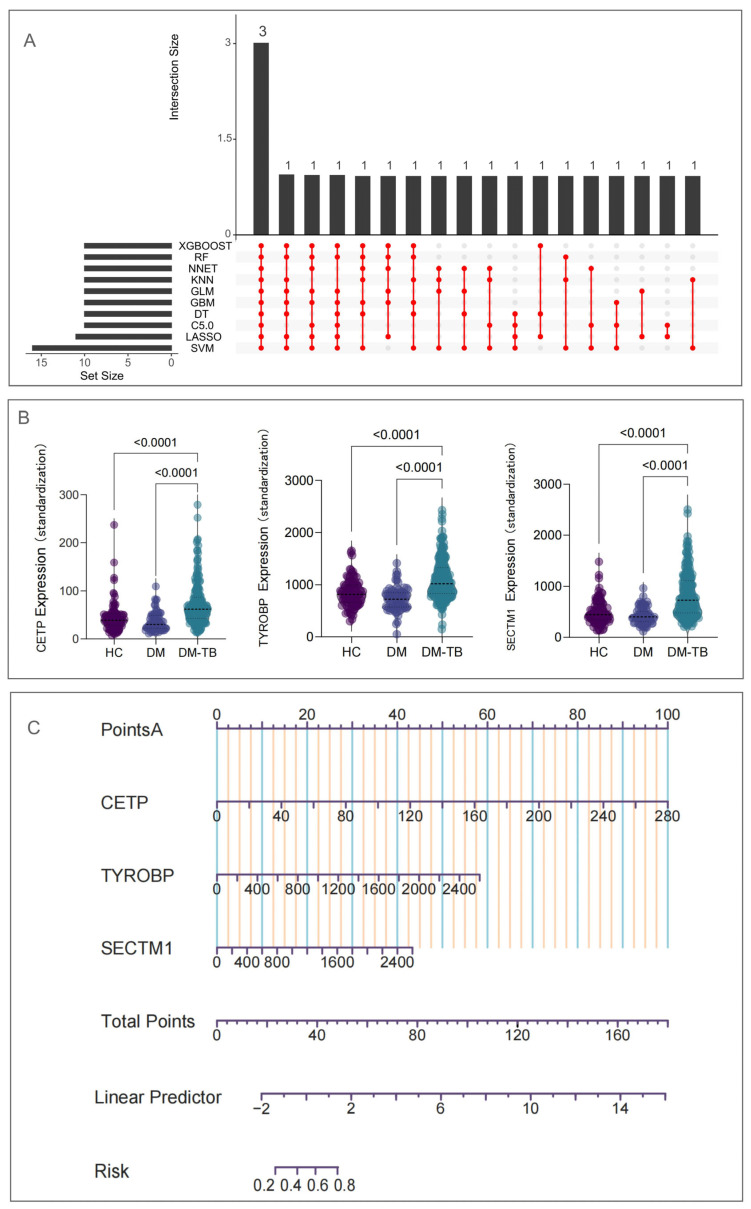
Selection of immune biomarkers and construction of the early risk alert model. (**A**) UpSet plot showing the overlap of immune core genes from 10 machine learning algorithms, resulting in 3 core genes. (**B**) Violin plot displaying the expression levels of the 3 genes in HCs, DM, and DM–TB groups, with Kruskal–Wallis non-parametric test comparing changes across the groups. (**C**) Visualization of the risk assessment model using a nomogram.

**Figure 8 microorganisms-13-00919-f008:**
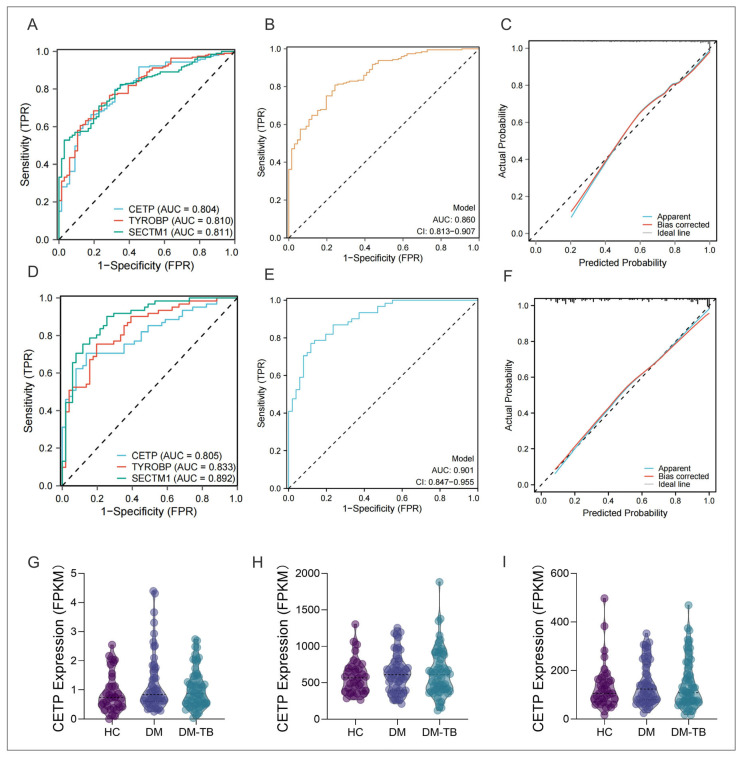
Evaluation of the biomarker and risk warning models of DM–TB in the dataset. (**A**–**F**) show the ROC curves and calibration curves of the models and the three genes in the GSE181143 training set (**A**–**C**) and the GSE114192 validation set (**D**–**F**), respectively. (**G**–**I**) show the expression level profiles of the three genes in the validation set.

**Figure 9 microorganisms-13-00919-f009:**
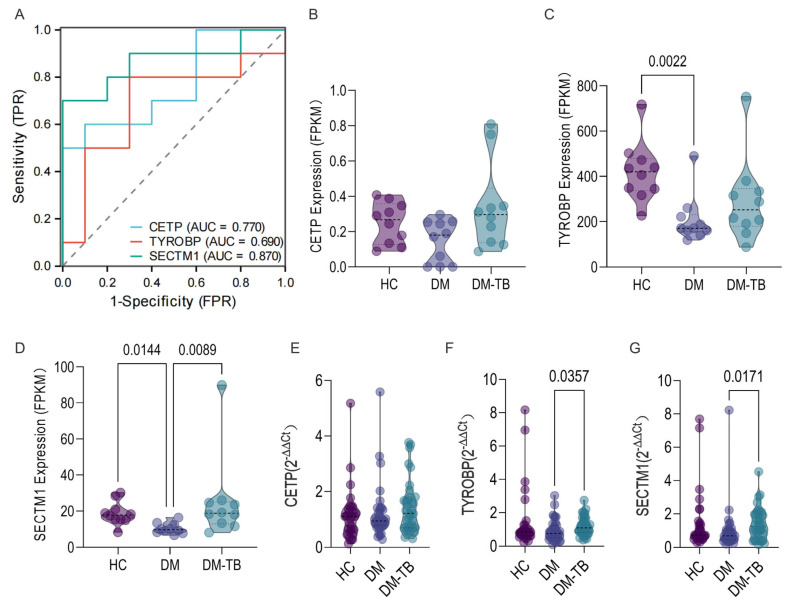
The validation of the biomarkers and risk warning model of DM–TB in a prospective cohort study and RT-qPCR. (**A**) shows the ROC curve analysis results of the three genes in the prospective cohort study. (**B**–**G**) show the expression of three genes in the prospective cohort study and RT-qPCR validation.

**Figure 10 microorganisms-13-00919-f010:**
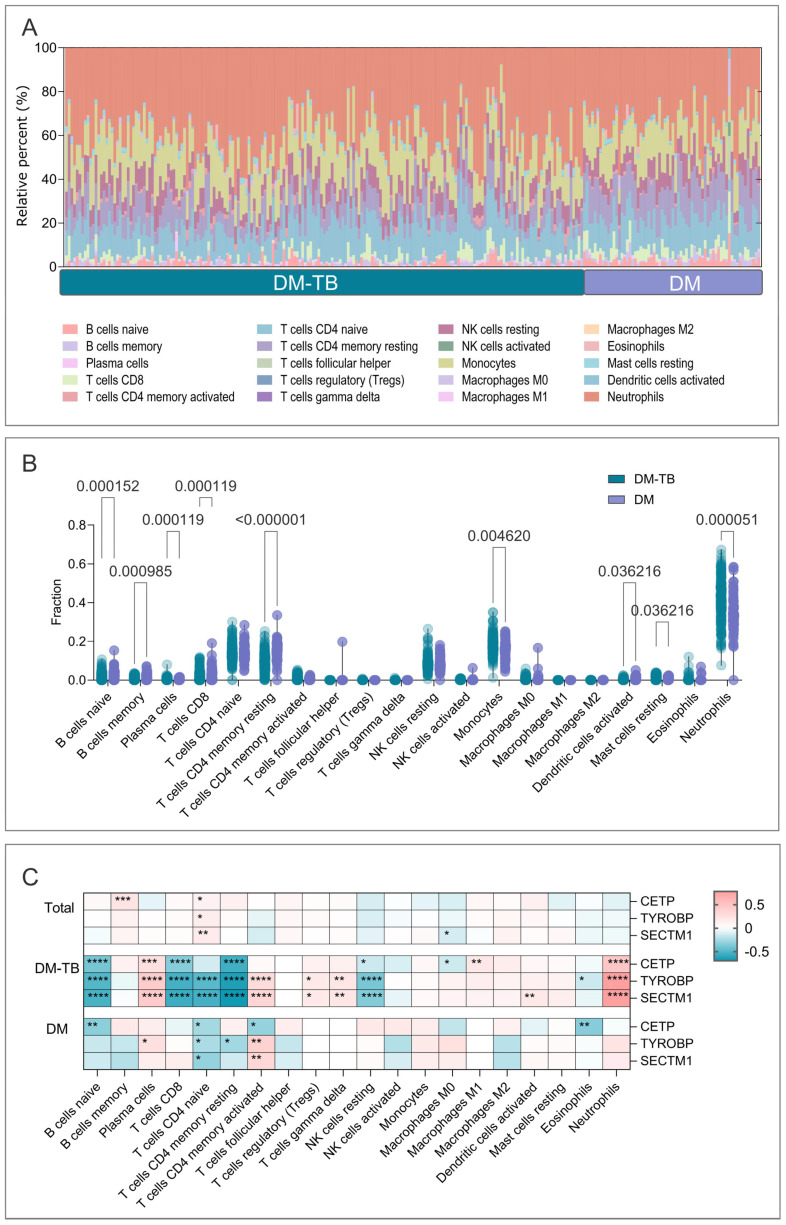
Immune infiltration analysis. (**A**) Immune cell infiltration levels in the DM–TB and DM groups. (**B**) Mann–Whitney test assessing changes in immune cell proportions between the two groups. (**C**) Correlation analysis between 3 genes and immune cells. *: indicate statistical significance: *: *p* < 0.05, **: *p* < 0.01, ***: *p* < 0.001, ****: *p* < 0.0001.

**Table 1 microorganisms-13-00919-t001:** Details of GEO data.

	Year	Accession	Platform	Sequencing Type	Sample (*n*)	Sample Source Country (*n*)	Species	Tissue
1	2021	GSE181143	GPL20795	RNA seq	201 (DM–TB: 71, DM: 55, HCs: 75)	Brazil (61) and India (140)	Homo sapiens	Whole blood
2	2020	GSE114192	GPL18573	RNA seq	149 (DM–TB: 61, DM: 52, HCs: 36)	Romania (44), Indonesia (19), South Africa (72), and Peru (12)	Homo sapiens	Whole blood

Abbreviations: DM, diabetes mellitus; DM–TB, coexistence of diabetes mellitus and tuberculosis; GEO, Gene Expression Omnibus; HCs, healthy controls; TB, tuberculosis.

**Table 2 microorganisms-13-00919-t002:** Detailed information on primer sequence of each gene symbol and internal reference gene.

Gene Symbol	Primer Sequence
CETP	Forward: 5′-GGCCAAGTCAAGTATGGGTTG-3′ Reverse: 5′-ACAGACACGTTCTGAATGGAGA-3′
TYROBP	Forward: 5′-ACTGAGACCGAGTCGCCTTAT-3′ Reverse: 5′-ATACGGCCTCTGTGTGTTGAG-3′
SECTM1	Forward: 5′-GGGACACCAGAGAAATAACAGACAAG-3′ Reverse: 5′-AGAGCGACCAAGAGGATGAAGAC-3′
GAPDH	Forward: 5′-CTCTGGTAAAGTGGATATTGT-3′ Reverse: 5′-GGTGGAATCATATTGGAACA-3′

**Table 3 microorganisms-13-00919-t003:** Comparison of clinical characteristics among (HCs, DM, and DM–TB cohorts from Brazil and India (GSE181143 dataset).

Variables *	HCs			DM			DM–TB		
	Brazil	India	*p*	Brazil	India	*p*	Brazil	India	*p*
N	15	60		15	40		31	40	
Age	35 (28–51)	35 (31–39)	0.64	56 (51–57)	52.5 (42–78)	0.46	48 (38–67)	48 (40–65)	0.7
Female no. (%)	9 (60%)	32 (53%)	0.6	8 (57%)	22 (55%)	0.88	9 (29%)	10 (25%)	0.7
BMI (kg/m^2^)	24.9 (20.5–28.9)	16.9 (16–20)	0.2	30.3 (26–32)	25.4 (23.6–27.7)	**<0.001**	22.5 (20–25.7)	21.9 (18–28.9)	0.79
Smoking (current)	5 (33.3%)	2 (3.3%)	**0.004**	3 (20%)	10 (25%)	0.6	12 (38.7%)	6 (15%)	0.02
Alcohol (current)	13 (86.7%)	11 (18.4%)	**<0.001**	14 (93.4%)	9 (22.5%)	**<0.001**	28 (90.3%)	5 (12.5%)	**<0.001**
Metformin	N/A	N/A	N/A	Not assessed	26 (65%)	Not assessed	6 (19.4%)	27 (87%)	**<0.001**
Statin	N/A	N/A	N/A	Not assessed	3 (7.5%)	Not assessed	Not assessed	9(22.5%)	**<0.001**
Cavitary TB	N/A	N/A	N/A	N/A	N/A	N/A	9 (29%)	26 (65%)	**<0.001**
HbA1c (%)	5.1 (4.9–5.2)	5 (5–5.5)	0.25	6.1 (5.9–7.4)	9.4 (8.4–11.1)	**<0.001**	8.5 (6.8–11.4)	11.7 (10–12.5)	**0.001**

*: Continuous variables (age, BMI, and HbA1c) are presented as medians with interquartile ranges (IQR), while categorical variables (female sex, current smoking, current alcohol consumption, metformin use, statin use, and cavitary tuberculosis) are reported as frequencies. Between-group comparisons were performed using the Kruskal–Wallis test (for non-parametric continuous variables) and the chi-square test (for categorical variables). Bolded *p*-values indicate statistical significance. This table was adapted from a previous study with permission [32], Copyright 2024, Elsevier (License Number 6001650294737). Abbreviations: BMI, Body Mass Index; HbA1c, glycated hemoglobin; N/A, Not Applicable.

**Table 4 microorganisms-13-00919-t004:** Logistic regression analysis of three hub DEGs.

Characteristics	OR	95% CI	*p*-Value
CETP	1.051	1.033–1.068	<0.001
TYROBP	1.004	1.003–1.006	<0.001
SECTM1	1.005	1.003–1.007	<0.001

## Data Availability

The original contributions presented in this study are included in the article/Appendix A. Further inquiries can be directed to the corresponding authors.

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
