# Peer review of "Development and Validation of Early Alert Model for Diabetes Mellitus–Tuberculosis Comorbidity"

_microorganisms, 2025, doi:10.3390/microorganisms13040919_

Round 1

Reviewer 1 Report

Comments and Suggestions for Authors

The manuscript titled "Development and Validation of an Early Alert Model for Diabetes Mellitus-Tuberculosis Comorbidity" presents a comprehensive and methodologically robust approach to addressing a significant global health challenge. The authors leverage advanced bioinformatics tools, machine learning algorithms, and transcriptomic data mining to identify immune-related biomarkers and construct an early alert model for patients with comorbid diabetes mellitus (DM) and tuberculosis (TB). This work is particularly noteworthy for its innovative integration of multi-omics data and its potential translational impact on clinical practice.

One of the standout advantages of the proposed approach lies in its use of weighted gene co-expression network analysis (WGCNA) combined with machine learning techniques. This dual-layered analytical framework allows for the identification of not only differentially expressed genes (DEGs) but also their functional relationships within biological networks. By focusing on co-expressed gene modules rather than isolated genes, the study captures the complexity of DM-TB pathogenesis more holistically. This systems biology perspective provides deeper insights into the molecular mechanisms underlying the disease, particularly the roles of NF-кB and MAPK signaling pathways, which are highlighted as central to the immune dysregulation observed in DM-TB patients.

Another significant strength of this work is the rigorous validation strategy employed by the authors. The early alert model was constructed using the GSE181143 dataset and subsequently validated using both the independent GSE114192 dataset and real-world patient cohorts. This multi-tiered validation process enhances the reliability and generalizability of the findings. Furthermore, the inclusion of RT-qPCR experiments to corroborate the expression levels of key biomarkers adds an additional layer of empirical support, bridging the gap between computational predictions and biological reality. Such thorough validation is often lacking in similar studies, making this manuscript a benchmark for future research in the field.

The choice of immune-related biomarkers – CETP, TYROBP, and SECTM1 – is another highlight of the study. These biomarkers were identified through a consensus-based approach that integrated multiple machine learning algorithms, ensuring their robustness and relevance. The high predictive performance of the early alert model, with an AUC of 0.86 in the training set and 0.901 in the validation set, underscores the potential clinical utility of these biomarkers. Moreover, the detailed immune infiltration analysis using the CIBERSORT algorithm provides valuable context by revealing shifts in immune cell populations, such as the suppression of adaptive immune cells and activation of inflammatory cells in DM-TB patients. This immunological characterization enriches the biological interpretation of the findings and opens new avenues for targeted therapeutic interventions.

From a mathematical and computational perspective, the study demonstrates exemplary rigor in its methodology. The use of ten distinct machine learning algorithms, including LASSO, Random Forests, and XGBoost, ensures a comprehensive exploration of the data. The application of the UpSet plot to visualize intersections among the top-ranked genes from different algorithms is particularly commendable, as it offers a clear and intuitive representation of consensus across models. Additionally, the authors' attention to statistical detail, such as the use of one-way ANOVA or Kruskal-Wallis tests based on data distribution, reflects a commitment to methodological precision.

The study presents a robust framework for early risk prediction in diabetes mellitus-tuberculosis (DM-TB) comorbidity, but there are several specific disadvantages and potential areas for improvement within the context of the research.

Authors used whole blood transcriptomics from GEO datasets, identified DEGs, did WGCNA and machine learning to find biomarkers, validated with cohorts and RT-qPCR, and performed immune infiltration analysis with CIBERSORT.

Looking for substantial cons related to the subject area. Let's start with the use of whole blood transcriptomics (Section 2.1). Bulk RNA-seq from whole blood can't distinguish cell-type-specific expression. For example, if a gene is upregulated in neutrophils but downregulated in lymphocytes, bulk data might show no change. This is critical because immune responses are cell-specific. The authors mention decreased CD8+ T cells in DM-TB (Figure 10B), but this could be due to fewer CD8+ cells rather than gene downregulation per cell. CIBERSORT's deconvolution (Section 3.8) has limitations in distinguishing similar cell types like M0/M1 macrophages. The authors should address this by suggesting cell-sorting or single-cell RNA-seq in future work.

Next, the inconsistency in CETP expression between training (GSE181143) and validation (GSE114192) datasets (Section 3.7). The authors attribute this to missing patient data, but it might reflect population-specific factors. For example, GSE114192 includes patients from Romania, Indonesia, etc., possibly with genetic differences affecting gene expression. The authors didn't adjust for covariates like ethnicity, age, or sex in their analysis (Section 2.10), which is a flaw. They should include covariate analysis to improve model generalizability.

In the RT-qPCR validation (Section 3.7), CETP didn't reach significance in the prospective cohort (Figure 9E). This discrepancy might arise because whole blood includes multiple cell types diluting the signal. If CETP is expressed in monocytes but not lymphocytes, using whole blood could mask changes. The authors should justify their choice of whole blood over isolated cells and discuss technical limitations.

The immune analysis paradox (Section 3.8): NF-кB/MAPK pathways are activated (Figure 4B), yet adaptive immunity is suppressed (Figure 10A). Typically, NF-кB activation promotes inflammation and T-cell responses. The observed suppression contradicts this, suggesting a unique mechanism in DM-TB comorbidity, like immune paralysis. The authors need to discuss this more deeply in Section 4, linking it to existing hypotheses about immune exhaustion in comorbidities.

The nomogram model (Figure 7C) uses a linear combination of biomarkers but ignores potential synergistic interactions. For example, CETP and SECTM1 might interact nonlinearly in lipid metabolism affecting macrophage function. The authors should explore nonlinear models like neural networks to capture such effects (Section 3.6).

The prospective cohort's small size (n=10 per group) underpowered CETP analysis (Figure 9B). Despite a decent AUC (0.77), the lack of significance questions CETP's reliability. The authors should revise conclusions about CETP in Section 4 and recommend larger studies.

Lastly, the study doesn't correlate biomarkers with clinical outcomes like TB severity or HbA1c levels (Table 1). Without linking biomarkers to disease progression or treatment response, their clinical utility is unclear. Adding clinical parameters to Table 1 would enable stratified analysis and enhance relevance.

The study addresses a highly relevant and original topic within the field of medical research, focusing on the development of an early alert model for diabetes mellitus-tuberculosis (DM-TB) comorbidity. This research is particularly timely, given the increasing global burden of both diabetes and tuberculosis, and the significant challenges posed by their co-occurrence.

Compared to other published materials, this study introduces a novel approach by integrating bioinformatics, machine learning, and comprehensive validation strategies to identify key biomarkers and develop a predictive model.

The conclusions drawn in the study are well-supported by the presented evidence and arguments. The use of multiple machine learning algorithms, external dataset validation, and real-world cohort studies ensures the robustness of the findings.

The references cited in the study appear relevant and up-to-date, providing a solid foundation for the research. They effectively support the methodologies employed and the interpretations of the results, ensuring that the study is well-grounded in the existing body of knowledge.

Regarding the tables and figures, they are generally well-presented and complement the text effectively. However, some figures, such as those depicting gene expression levels and immune cell infiltration, could benefit from clearer annotations and legends to enhance interpretability. Additionally, ensuring that all abbreviations are defined, either in the figure legends or within the main text, would improve accessibility for readers who may not be familiar with the specialized terminology.

In conclusion, this manuscript represents a significant advancement in the field of DM-TB comorbidity research. Its strengths lie in the innovative combination of WGCNA and machine learning, the meticulous validation process, and the identification of clinically relevant biomarkers. The study not only enhances our understanding of the molecular and immunological underpinnings of DM-TB but also provides a practical tool for early risk prediction. Such an approach has the potential to transform clinical management strategies, ultimately improving patient outcomes and reducing the global burden of these intertwined diseases.

Author Response

Responses to reviewer 1:

  1. Figures must be more clear, there are difficult to read.

Response: Thank you for your feedback. We recognize that some of the charts lack sufficient resolution and clarity. We have revised some of the diagrams by increasing the resolution, adjusting the font size to improve readability, and adding clearer annotations. High-resolution versions of all charts will be submitted with the revisions.

  1. Why authors used only these databases. Genes might be different in other races , that's why their results can not generalized.

Response: In addition to the GEO database, we pre-searched the ArrayExpress, EBI European Nucleotide Archive and TB Portals databases and screened them according to strict inclusion criteria (whole blood RNA-seq, case-control design, sample size ≥3 per group), and the final eligible ones were the 2 GEO datasets ( GSE181143 and GSE114192). Although these datasets include diverse populations (e.g., Brazil, India, Romania, Indonesia), we recognize that genetic and environmental factors may influence gene expression. For this reason, we have added a discussion of the need for validation in multi-ethnic cohorts in Section 4 to emphasize the importance of population-specific studies. Future work will prioritize the inclusion of broader demographics. Now, it is read as: “Fourth, the cohort consisted mainly of populations from specific regions (e.g., Brazil and India) and lacked validation in multiethnic groups. Genetic and environmental heterogeneity across populations may affect biomarker performance, necessitating ex-ternal validation across populations.” (Lines 668-671).

  1. Although authors mentioned that they found some genes and biomarkers that can predictthese entities, they do not mention how this could be helpful in general practice and how thisaffects this screening the economic aspects.

Response: We appreciate this key point. In the revised Discussion section, we have added a discussion on the potential clinical utility of CETP, TYROBP, and SECTM1 as cost-effective biomarkers for early screening of DM-TB. Now, it is read as: “The identified biomarkers (CETP, TYROBP, SECTM1) hold translational potential for non-invasive TB screening in diabetic populations.  Their detection via cost-effective RT-qPCR could enable early risk stratification in high-burden regions, reducing reliance on costly radiologic or pathogen-based diagnostics. By facilitating timely prophylaxis, this approach may curb hospitalization rates, mitigate drug resistance, and lower long-term healthcare costs.” (Line 653-657).

  1. Authors used databases, however their results could be different if they used patients samples.

Response: We agree that database-derived results require real-world validation. Our study combined prospective and retrospective cohorts (Sections 2.7-2.8) to validate gene expression trends, and it turned out that specific markers differed in our cohort. We speculate that technical constraints such as batch effects and cell type heterogeneity in whole blood samples may be responsible. To mitigate these challenges, we recommend the use of isolated cell populations (e.g., PBMCs) in future studies to minimize technical variability and improve the reproducibility of biomarkers in different populations. We have added this point into our limitations in the revised manuscript, now it is read as: “Finally, bulk RNA-seq of whole blood inherently masks cell type-specific expression patterns due to the heterogeneous cellular composition of peripheral blood. Future studies should prioritize the integration of single-cell RNA sequencing (scRNA-seq) or fluorescence-activated cell sorting (FACS), which can pinpoint biomarker expression in disease-associated immune subpopulations (e.g., monocytes, neutrophils).” (Line 674-679).

Reviewer 2 Report

Comments and Suggestions for Authors

A very i teresting study, although there so e things need improvement

  1. Figures must be more clear, there are difficult to read
  2. Why authors used only these databases. Genes might be different in other races , that's why their results can not generalized
  3. Although authors mentioned that they found some genes and biomarkers that can predict these entities, they do not mention how this could be helpful in general practice and how this affects this screening the economic aspects
  4. Authors used databases, however their results could be different if they used patients samples

Author Response

Please see the Word file for my responses to Reviewer 2.
